# Effects of Antiarrhythmic Drugs on Antiepileptic Drug Action—A Critical Review of Experimental Findings

**DOI:** 10.3390/ijms23052891

**Published:** 2022-03-07

**Authors:** Kinga K. Borowicz-Reutt

**Affiliations:** Independent Unit of Experimental Neuropathophysiology, Department of Toxicology, Medical University of Lublin, 20-090 Lublin, Poland; kingaborowicz@umlub.pl

**Keywords:** antiarrhythmic drugs, antiepileptic drugs, interactions, maximal electroshock

## Abstract

Severe cardiac arrhythmias developing in the course of seizures increase the risk of SUDEP (sudden unexpected death in epilepsy). Hence, epilepsy patients with pre-existing arrhythmias should receive appropriate pharmacotherapy. Concomitant treatment with antiarrhythmic and antiseizure medications creates, however, the possibility of drug–drug interactions. This is due, among other reasons, to a similar mechanism of action. Both groups of drugs inhibit the conduction of electrical impulses in excitable tissues. The aim of this review was the analysis of such interactions in animal seizure models, including the maximal electroshock (MES) test in mice, a widely accepted screening test for antiepileptic drugs.

## 1. Introduction

According to WHO, epilepsy is the second most common neurological disorder globally. Around 65 million people worldwide suffer from this disease. Since Western populations are aging fast and the incidence of epilepsy increases with age, the prevalence of this condition is rising constantly. The first written document about epilepsy was found in Mesopotamia. Back in antiquity and the Middle Ages, seizures were thought to result from possession and contributed to the condemnation of innocent people for witchcraft. Nowadays, nobody associates epilepsy with the action of demons. However, many myths have arisen around the disease and numerous patients still feel stigmatized and discriminated. Furthermore, patients with drug-resistant epilepsy cannot live fully fledged lives, as they are unable to fulfill their professional and social roles. The cumulative effects of recurring seizures lead to an increased rate of marital and family breakdown, unemployment, impaired career progress, and consequent financial difficulties [1]. Moreover, drug-resistant patients have an increased risk of SUDEP (sudden unexpected death in epilepsy) development. The exact pathophysiology of SUDEP is currently unknown, although it is believed to be related to frequent incidence of generalized tonic–clonic convulsions and the resulting cardiac, respiratory, and brainstem disorders in the mechanism affecting the function of the central autonomic control centers [2,3]. This was one of the reasons why the maximal electroshock (MES) test was selected in this study. It is the most commonly used animal model reflecting generalized tonic–clonic seizures in humans.

In up to even 42% of cases, epilepsy coexists with arrhythmias, the most frequent one being atrial fibrillation, sudden cardiac arrest, bundle branch block, and ventricular tachycardia [4]. This co-occurrence may result from the common genetic background of the two disorders, e.g., mutation of genes encoding Na^+^, K^+^, and Ca^2+^ ion channels [5,6]. Patients with long-lasting epilepsy often present some interictal cardiac changes, including QT prolongation, decreased heart rate variability, subtle signs of ischemia (ST-segment depression), and ventricular late potentials [7]. Importantly, significant prolongation of QT, probably due to the altered function of sodium channels, was observed in patients who experienced SUDEP [5,8]. Experimental research provided evidence that seizures can disturb autonomic regulation of the heart and lead to fatal arrhythmias [9]. In turn, clinical studies showed that seizures may be preceded by tachycardia or atrial/ventricular ectopy [8].

Epilepsy patients with serious arrhythmias should be treated with antiseizure and antiarrhythmic drugs simultaneously. Either classical or newer antiepileptic drugs are also used in the treatment of disorders other than epilepsy, including bipolar disorders (valproate, carbamazepine, oxcarbazepine, lamotrigine), migraine (topiramate, lamotrigine), neuropathic pain (valproate, carbamazepine, lamotrigine, oxcarbazepine, pregabalin), and fibromyalgia (pregabalin) [10]. Such a wide range of applications increases the likelihood of polypragmasy with antiseizure and antiarrhythmic drugs, which considerably rises the risk of interactions between them. The two groups of medications present similar mechanisms of action, modulating the function of ion channels. There are also similarities between cardiac and neural action potentials. This creates the theoretical basis for antiarythmogenic properties of antiepileptic drugs and anticonvulsant effects of antiarrhythmic medications [4,11,12]. Interestingly, phenytoin is classified as both an antiseizure and antiarrhythmic drug. On the other hand, lidocaine has been effective in certain cases of drug-resistant seizures. It should be underlined, however, that “the dose makes the poison”. Antiepileptics in overdose can be arrhythmogenic, On the other hand, antiarrhytmic drugs (particularly sodium channel blockers) applied at supratherapeutic doses may, though rarely, generate seizures [4]. Furthermore, some antiepileptic drugs induce the hepatic microsomal system, which decreases the effectiveness of the concomitant antiarrhythmic treatment [13].

According to Vaughan Williams, antiarrhythmic drugs are classified into four groups [4]. Class I includes moderate (IA), weak (IB), or marked (IC) sodium channel blockers, which reduce action potential phase 0 slope and overshoot while increasing, decreasing or preserving the action potential duration and effective refractory period, respectively. The resulting reduction in the excitability of cardiomyocytes suppresses abnormal rhythms. Since sinoatrial and atrioventricular nodes use calcium ions to depolarize, class I antiarrhythmic have a negligible effect on the pacemaker cells. Hence, sodium channel blockers can be used in re-entry tachyarrhythmias, in which a blockade of the atrioventricular node is unfavorable. A representative of class IA is procainamide. Class IB comprises lidocaine, mexiletine, and phenytoin, while propafenone and flecainide belong to class IC. Class II contains β-adrenergic blockers that reduce action potential phase 4 slope, decrease sinoatrial node pacing rates, and slow atrioventricular node conduction. B-blockers differ from each other in terms of β1/β2 receptor selectivity, intrinsic sympathomimetic activity, and membrane-stabilizing (local anesthetic) activity. Examples of class II antiarrhythmics are propranolol, metoprolol, atenolol, esmolol, and timolol. Class III, comprising potassium channel blockers, inhibit action potential phase 3 repolarization (potassium efflux) and lengthen the effective refractory period. Representatives of this class are: amiodarone, bretylium, dofetilide, ibutilide, and sotalol. Sotalol, in addition to class III properties, is also a nonselective β-blocker without intrinsic sympathomimetic or membrane-stabilizing activity. Finally, class IV drugs, containing calcium channel blockers, reduce action potential phase 0 in the sinoatrial and atrioventricular nodes, decreasing heart rate and conduction. Calcium blockers also inhibit action potential phase 2 in cardiomyocytes and suppress contraction. The main representatives of class IV antiarrhythmics are verapamil and diltiazem [4,14,15] (Figure 1).

Recently, several other classes have been added to this classification. Class ID relates to actions on late sodium current (INaL) components important in long QT syndrome type 3. Class II takes into account advances in the understanding of autonomic, often G protein-mediated, signaling. Class III covers numerous subsequently discovered potassium channels regulating action potential and refractory period durations. Class IV comprises not only calcium channel blockers, but also drugs that modify intracellular calcium homeostasis. Further new classes have been added, such as class 0, taking into account cardiac automaticity, class V, including drugs acting on mechanically sensitive channels, class VI, gathering factors affecting electrotonic coupling between cells, and class VII, containing substances modifying structural remodeling (class VII) [15]. The only representative of class 0 is ivabradine, which slows the heart rate by inhibiting *If*/*Ih* currents conducted through hyperpolarization-activated cyclic nucleotide-gated (HCN) channels. Other considered mechanisms of action include deactivation of outward delayed rectifier potassium current and activation of inward currents, such as sodium-dependent background currents (IbNa), T- and L-type calcium currents (ICaL and ICaT), and sustained inward currents (Ist) [15]. *If* (“funny current”) occurs in the sinoatrial node, while corresponding *Ih* currents are observed in neurons. Both of them are described as voltage-activated Na^+^/K^+^ hyperpolarization-activated depolarizing currents facilitated by cAMP. To date, four isoforms of HCN have been identified. In the heart, *If* is responsible for initiation and regulation of the heart beat (“pacemaker current”). Roles attributed to *Ih* in the nervous tissue include control of rhythmic activity in neuronal circuits (e.g., in the thalamus), regulation of excitability, determination of resting membrane potential, dendritic integration, and synaptic transmission. Therefore, dysfunctions of HCN channels may be involved in some forms of epileptic activity [10,16,17,18], and ivabradine has a theoretical background in interacting with antiseizure drugs.

Detailed characteristics of remaining new classes of antiarrhythmics are far beyond the scope of this review (for further information see Lei et al. [15]).

Summing up, the purpose of this review was to analyze the influence of antiarrhythmic drugs and ivabradine on the effectiveness of antiseizure treatment. Effects of antiarrhythmic drugs on seizure phenomena depend on many variables, including the drug dose, route, frequency, and duration of drug administration as well as the seizure model and animal species used in experiments. To avoid significant variability in results, this review has been limited to studies conducted in the maximal electroshock (MES) test in mice, the most common screening model used in the search for potential antiepileptic drugs. Detailed pharmacokinetic considerations were intentionally omitted, since pharmacokinetics of drugs in mice and humans may markedly differ and results are not always extrapolatable.

## 2. Selection of Literature

Medline and Science Direct databases were searched for additional articles using the term “antiarrhythmic drugs and maximal electroshock”. Out of 74 results found in Medline and Science Direct, 23 articles were selected as the most relevant to the search criteria. These articles contain the results of research conducted over the past 30 years.

## 3. Antiarrhythmic Drugs Have Different Effects on Tonic–Clonic Convulsions

In most articles included in this review, the MES parameters were as follows: 25 mA, 50 Hz, and 0.2 s. In remaining cases, MES details and their results were described together. Results were considered significant when *p* was lower than 0.05.

### 3.1. Class 0 Antiarrhythmic Drugs

Ivabradine administered at doses of 5 and 10 mg/kg did not alter the electroconvulsive threshold in mice; however, at higher doses of 15 and 20 mg/kg, it significantly elevated this parameter. Ivabradine (10 mg/kg) significantly enhanced the antielectroshock activity of valproate and reduced the action of phenytoin and lamotrigine in the MES test in mice. The effectiveness of carbamazepine, phenobarbital, lacosamide, pregabalin, and topiramate remained unchanged. Furthermore, ivabradine significantly diminished the brain concentration of phenytoin and had no effect on the brain levels of remaining antiepileptic drugs used in this study. This suggests that the interaction between ivabradine and phenytoin is at least partially due to pharmacokinetic events, while the interaction between ivabradine and valproate seems to be pharmacodynamic [19,20,21].

### 3.2. Class I Antiarrhythmic Drugs

Propafenone at the dose range of 60–90 mg/kg significantly elevated the electroconvulsive threshold (ECT), being ineffective at lower doses. Applied at its subthreshold dosages, propafenone potentiated the antielectroshock action of seven antiepileptic drugs: carbamazepine, valproate, phenytoin, phenobarbital, oxcarbazepine, topiramate, and pregabalin. The action of lamotrigine remained unaffected. Interactions between propafenone and valproate or pregabalin may be in part due to pharmacokinetic events, since propafenone significantly elevated the brain levels of the two antiepileptics. On the other hand, propafenone potentiated the action of carbamazepine despite its lowered concentration in the brain. It may suggest actual synergism between propafenone and carbamazepine, which is, however, masked by pharmacokinetic interactions [22,23].

Mexiletine exhibited properties of an antiseizure drug, being active not only in the ECT but also in the MES test. This enabled isobolographic analysis of interactions between mexiletine and antiepileptic drugs. Regarding classical antiepileptics, antagonistic interaction was revealed between mexiletine and valproate for two fixed-ratio combinations of 1:1 and 3:1. Additivity was observed between mexiletine and valproate applied in the proportion of 1:3. Moreover, mexiletine interacted additively with carbamazepine, phenytoin, and phenobarbital in all three fixed ratios. Since mexiletine did not significantly alter brain concentrations of carbamazepine, phenobarbital, and phenytoin, the observed interactions seem to be pharmacodynamic in nature. In contrast, the antiarrhythmic drug decreased the brain level of valproate. This could be, at least in part, the reason for antagonistic interaction between the two drugs [24].

In relation to new-generation antiepileptic drugs, the mixture of mexiletine and pregabalin at the fixed ratios of 1:1 and 3:1 led to synergistic interaction, while the combination in the proportion of 1:3 was additive. Synergism was also demonstrated for the combination of mexiletine with topiramate in all three proportions. Combinations of this antiarrhythmic drug with lamotrigine or oxcarbazepine were found to be additive. Interestingly, synergism between mexiletine and topiramate at the dose ratio of 1:1 existed despite the mexiletine-induced decrease in the brain concentration of topiramate. This may indicate strong pharmacodynamic interaction between the two drugs. A similar situation occurred with the combination of mexiletine and oxcarbazepine (1:1) as well as pregabalin (1:3), where additivity was enough to overcome decreased brain levels of the two antiepileptics. It may suggest that in this case, pharmacokinetic interactions mask the actual synergism between above-mentioned drugs. It is worth underlining that pharmacokinetic events may vary depending on the proportion of combined drugs. Mexiletine decreased the brain concentration of pregabalin at the dose ratio of 1:3, increased this level at 1:1, and did not alter it in the proportion of 3:1 [25].

### 3.3. Class II Antiarrhythmic Drugs

In one of the first studies on this topic, propranolol applied at doses of 5, 10, and 20 mg/kg increased the ECT in mice in a dose-dependent manner. According to the authors, the anticonvulsant action of this β-blocker is related to its membrane-stabilizing properties [26]. Moreover, propranolol (40 mg/kg) showed antielectroshock activity comparable to that of phenytoin (30 mg/kg). Parameters of the MES in this study were: 60-Hz current of 50 mA intensity for 0.2 s through ear-clip electrodes [27]. In the study conducted by Fischer and Müller [28], some β-blockers with local anesthetic properties (propranolol, alprenolol, and pindolol, all at 10 mg/kg) increased the antielectroshock action of phenobarbital. The MES was provided with following parameters: 35-Hz current of 50 mA intensity for 0.4 s through ear-clip electrodes. In another study, (+/−) propranolol (1–50 mg/kg), (+) propranolol (50 mg/kg), and pindolol (10–50 mg/kg) exhibited significant protective effects against the MES (30 mA, 0.2 s, no data about current frequency), whereas timolol (1 mg/kg), and propranolol analog UM-272 (1 and 10 mg/kg) were ineffective in this respect [29].

In another study, propranolol, acebutolol, metoprolol, and atenolol potentiated the antiseizure action of certain antiepileptic drugs against the MES test in mice. Propranolol and metoprolol increased the effectiveness of valproate and diazepam. Acebutolol enhanced the action of valproate but not that of diazepam. In contrast, atenolol, which does not penetrate the blood–brain barrier, had no effect on the two antiepileptic drugs. None of the β-blockers changed the protective activity of carbamazepine and phenytoin against the MES. Revealed interactions do not seem to be pharmacokinetic, since β-blockers did not change the brain levels of valproate or diazepam. Propranolol and metoprolol are highly lipophilic agents, easily penetrating to the brain, whereas acebutolol crosses the blood–brain barrier to a moderate degree. Hence, it may be suggested that the action of separate beta-blockers on the action of antiepileptic drugs depends largely on their brain levels [30]. Some β-receptor blockers with local anesthetic properties (e.g., propranolol 5–10 mg/kg, alprenolol 10 mg/kg, pindolol 10 mg/kg, all at the dose of 10 mg/kg) were able to enhance the protective effect of phenobarbital in higher concentrations [28].

Finally, it was shown that nebivolol (0.5–15 mg/kg) did not raise the ECT but, at the dose of 15 mg/kg, it reduced the antielectroshock properties of carbamazepine. The effect of valproate, phenytoin, and phenobarbital remained unchanged by this β-blocker. Nebivolol significantly decreased the brain concentration of valproate but did not affect concentrations of remaining antiepileptic drugs [31].

### 3.4. Class III Antiarrhythmic Drugs

Although amiodarone (25–75 mg/kg) did not change the ECT, when applied at the dose of 75 mg/kg, it significantly enhanced the antielectroshock activity of carbamazepine, oxcarbazepine, and pregabalin in mice. The action of valproate, phenytoin, phenobarbital, lamotrigine, and topiramate remained unaffected. Brain concentrations of antiepileptic drugs were not affected by amiodarone. Therefore, the interaction between amiodarone and carbamazepine also seems to be pharmacodynamic in nature [32,33].

Dronedarone, another multichannel blocker, administered alone (in doses of 50, 75, and 100 mg/kg), increased the ECT in mice. Surprisingly, this amiodarone derivative (50 mg/kg) significantly reduced the anticonvulsant action of phenytoin in the MES test. No effect on the action of carbamazepine, phenobarbital, or valproate was observed [34]. Dronedarone (50 mg/kg) significantly enhanced the anticonvulsant potency of lamotrigine but did not affect the anticonvulsant properties of lacosamide, pregabalin, or topiramate in the MES test in mice. The measurement of total brain concentrations of phenytoin and lamotrigine revealed that dronedarone did not significantly alter total brain concentrations of lamotrigine in experimental animals, so interactions observed between dronedarone and the two antiepileptics seem to be pharmacodynamic in nature [35].

Sotalol at doses up to 100 mg/kg did not affect the ECT. This antiarrhytmic drug applied at the dose range of 60–100 mg/kg potentiated the antielectroshock action of valproate, whilst it potentiated that of phenytoin at doses of 80–100 mg/kg. Sotalol did not, however, affect the action of carbamazepine, phenobarbital, oxcarbazepine, lamotrigine, pregabalin, or topiramate in the MES test. Furthermore, sotalol significantly decreased the brain concentration of lamotrigine, increased those of oxcarbazepine and topiramate, and did not change the levels of remaining antiepileptic drugs. This indicates that interactions between sotalol and valproate as well as sotalol and phenytoin are most likely pharmacodynamic [36,37].

### 3.5. Class IV Antiarrhythmic Drugs

In the study conducted by Czuczwar et al. [38], diltiazem elevated the ECT at doses of 2.5 and 5 mg/kg, but at doses of 1.25 and 10 mg/kg, it did not affect this parameter. This calcium blocker at the dose of 1.25 mg/kg markedly potentiated the protective action of carbamazepine and diphenylhydantoin against the MES-induced seizures in mice. When applied at the higher dose of 2.5 mg/kg, it also enhanced the action of phenobarbital and valproate [38].

In newer reports, diltiazem (up to 10 mg/kg, ip) and verapamil (up to 20 mg/kg, ip) did not significantly affect the ECT in mice. Diltiazem (5 and 10 mg/kg) markedly potentiated the antielectroshock activity of topiramate, but not that of lamotrigine. In contrast, verapamil (5, 10, and 20 mg/kg) had no effect on the antiseizure action of topiramate. Pharmacokinetic verification revealed that diltiazem did not affect the brain concentration of topiramate or lamotrigine; thus, the observed interactions were considered as pharmacodynamic in nature [39,40].

All data outlines can be found in Table 1, Table 2 and Table 3.

### 3.6. Clinical Data

Clinical knowledge of the influence of antiarrhythmic drugs on seizures is very scarce. Lidocaine and propranolol were effective in patients with chronically unstable generalized epilepsy [41,42]. In 3 of 11 patients, propranolol decreased, by at least 50%, the frequency of startle-induced epileptic seizures [43]. Mexiletine and lidocaine attenuated seizures in symptomatic partial epilepsy and Lennox–Gastaut syndrome and controlled neonatal seizures resistant to phenobarbital, lamotrigine, and midazolam [44,45,46]. On the other hand, verapamil limited convulsions in recurrent status epilepticus [47,48].

## 4. Discussion

### 4.1. Probable Mechanisms of the Anticonvulsant Action of Antiarrhythmic Drugs

In light of the above data, the blockade of sodium channels seems to be the most important in the mechanism of the antiseizure action of antiarrhythmic drugs. Inhibition of potassium and calcium currents appears to be an additional factor contributing to such an effect. Voltage-dependent (gated) sodium channels (VDSCs) are crucial in the initiation and propagation of electrical signals in excitable tissues. Therefore, their blockade was effective in the treatment of epilepsy, cardiac arrhythmias, and chronic pain [49,50].

However, the VDSC family is quite diverse and has nine members, from Nav1.1 to Nav1.9. As is widely reported elsewhere, mutations of Nav1.1 and Nav1.2 are related to epilepsy, whilst Nav1.5 mutations have a significant relation to cardiac arrhythmias. Nav channels consist of a pore-forming α-subunit composed of four domains (I, II, III, and IV) connected through intracellular linkers, which can be coupled to one or two auxiliary β-subunits. Each domain of the α-subunit consists of six transmembrane helical segments (S1-6) [50,51]. Within domain II, there is a pore module (PM) composed of S5 and S6 segments and the intervening P loop, as well as the voltage-sensing module (VS) created by S1–S4 segments. S1-S6 segments in domain III serve as the fast inactivation gate. In a cross section of the channel, we can see a large external vestibule, a narrow ion selectivity filter (SF), a large central cavity lined by S6 segments, and an intracellular activation gate [51,52].

From a physiological point of view, VDSCs exist in three functional states: open, closed (resting), and inactivated. During depolarization, the voltage sensors move outward, and the pore opens. Then, fast and slow inactivation turns the channel into a non-conducting inactivated state. It was reported that various VDSC blockers may have different affinities for each state. Some of them present so-called use dependence, which means that the compound potency increases with a higher frequency of channel opening. VDSCs are blocked by drugs used clinically as local anesthetics, antiarrhythmics, and antiepileptics. These drugs are largely unselective within the channel family, which may contribute to potential undesirable effects limiting their application. Aforementioned substances bind to the so-called local anesthetic binding site located within the channel pore and formed by S6 segments in domains I, III, and IV. In detail, flecainide was reported to bind to a site in the central cavity of Nav1.5, just on the intracellular side of the selectivity filter [49]. Access to this site requires, however, opening of the intracellular activation gate, which constitutes the structural basis for the use-dependency phenomenon [51,53].

#### 4.1.1. Class 0 Antiarrhythmic Drugs

Ivabradine, the inhibitor of *If*/*Ih* currents, blocks HCN channels [16]. In addition to its negative chronotropic effect, this drug also presents anticonvulsant, antioxidant, and neuroprotective effects. The molecular docking technique revealed a high affinity of ivabradine to GABA_A_ receptors, which may explain the anticonvulsant action of the drug in the ECT test. On the other hand, no affinity to NMDA receptors was shown [54]. Since HCN channels are known to be involved in generation of absence seizures, it is not surprising that ivabradine inhibited experimental absence seizures in rats [55]. This drug also increased the ECT in mice.

Experiments performed in the model of pentetrazole-induced convulsions demonstrated that the blockade of HCN_1_ channels does not affect convulsions and inhibition of the HCN_2_ type increases seizure susceptibility, while suppression of HCN_4_ channels leads to an antiseizure effect [56]. These findings are in line with earlier observation that HCN_2_-deficient mice exhibited spontaneous absence seizures [57]. In light of the above data, it is surprising that ivabradine blocks HCN_2_ channels. It is possible other, unknown mechanisms determine the action of this drug in the ECT.

#### 4.1.2. Class I Antiarrhythmic Drugs

Undoubtedly, class I antiarrhytmics appeared the most effective of all groups with regard to the antiseizure activity. Therefore, it can be assumed that the Nav channel blockade is crucial in this process. Both propafenone and mexiletine block fast Nav1.5 channels and KCNH_2_ potassium channels. In addition, both of them present membrane-stabilizing effects [58]. Furthermore, propafenone showed β-sympatholytic activity at about 1/50 of the potency of propranolol [2,59]. All these pharmacological properties may contribute to antiseizure effects of the class I antiarrhythmics in the ECT test in mice.

#### 4.1.3. Class II Antiarrhythmic Drugs

The hippocampus, the brain structure strongly involved in generation and propagation of seizures [60], has the highest β_1_/β_2_ receptors among all brain structures [61]. However, the role of β-receptors in epileptic processes is equivocal, since their stimulation showed either pro- or anticonvulsant effects. Among 8 β-blockers compared in this review, only propranolol and pindolol (both given at the dose of 50 mg/kg) attenuated the MES-induced convulsions in mice. The parameters were: 50 mA, 35 Hz, and 0.4 s [28]. Therefore, it is clear that the two β-blockers increase the ECT at lower doses.

Propranolol is a nonselective, highly lipophilic, and membrane-stabilizing β-blocker. Pindolol is a nonselective, β adrenoceptor antagonist with substantial intrinsic sympathomimetic (partial agonist) activity. Pindolol has low membrane-stabilizing activity and moderate lipid solubility [62]. The common properties of the two drugs are lipophilicity, nonselectivity, and cell membrane stabilization (see Table 4). Antiseizure properties of β-blockers may be proportional to their lipophilicity and, hence, their permeability through the blood–brain barrier. However, it is not enough to explain their antiseizure action in the ECT, since metoprolol, as lipophilic as pindolol, did not show any activity in this test. Another mechanism related to the blockade of β-receptors is a decrease in the brain levels of cAMP. This in turn reduces activity of glutamatergic neurotransmission [30]. In addition, timolol shares all above-mentioned properties of propranolol and pindolol but did not affect the ECT.

Nebivolol is a highly selective β-blocker with membrane-stabilizing activity. The uniqueness of nebivolol is related to its agonistic interaction with β_3_ adrenoceptors and subsequent stimulation of endothelial nitric oxide synthase. Consequent nitric-oxide-mediated vasodilation is a reason why nebivolol is recommended for the treatment of hypertension. Although β_3_ receptors are the most widespread in the adipose tissue, their presence was also found in the brain, with the highest density in the hippocampus, cortex, and striatum. Some β_3_ agonists were reported to be effective in animal models of depression and anxiety. The lack of activity in the ECT suggests that β_3_ receptor stimulation is not relevant in antiseizure effects [31].

#### 4.1.4. Class III Antiarrhythmic Drugs

The only drug from this group that increased the ECT was dronedarone. Amiodarone and dronedarone are known as multichannel blockers, showing an affinity for potassium, sodium, and calcium channels. Both antiarrhythmics suppress conductivity through L-type calcium channels and voltage-gated channel subfamily H member 2 potassium channels (KCNH_2_). KCNH_2_ channels suppress the influx of K^+^ ions during repolarization and may be a potential link between epilepsy and arrhythmias related to long QT syndrome [65]. Moreover, amiodarone preferentially inhibits the late over transient sodium current; however, this effect was significantly weaker than the blockade of potassium channels. Apart from KCNH_2_, amiodarone suppresses other outward potassium currents: the rapidly and slowly activating delayed-rectifier potassium currents (iKr, iKs) and acetylcholine-activated potassium currents (iKAch, iKNa, iKATP). The blockade of calcium channels also involves their N, P, Q, and T types. In addition, amiodarone noncompetitively inhibits β_1_ and β_2_ adrenoceptors, as well as block current carried through HCN_2_ and HCN_4_ channels [66,67,68,69] (DrugBank, https://go.drugbank.com (accessed on 10 February 2022)). An antiarrhythmic-depressed calcium-dependent release of glutamate was also reported [70]. Among all these mechanisms, the blockade of potassium channels may be a proconvulsive factor, whereas remaining ones are largely associated with an anticonvulsant effect. The net effect on the ECT was, however, neutral.

Dronedarone, another multichannel blocker and amiodarone derivative, expresses analogous action on ion channels to the parent drug. The main difference is that dronedarone is an antagonist of not only β but also of α-receptors [71]. However, adrenolytics were not reported to affect the seizure threshold [29]. Therefore, the reason for the antiseizure action of dronedarone remains unclear. It can be only speculated that the weaker blockade of potassium channels than in the case of amiodarone contributes to this effect.

Sotalol is a medication sharing properties of class III and class II antiarrhythmic drugs. Sotalol, a racemic 1:1 mixture of D- and L-isomers, is a nonselective competitive β-blocker without intrinsic sympathomimetic or membrane-stabilizing activity. Interestingly, both isomers lengthen the cardiac repolarization phase, but the blockade of β-receptors is almost entirely caused by L-isomer. The latter effect prevents production of cAMP and decreases calcium influx. Similarly to amiodarone and dronedarone, sotalol blocks KCNH_2_ receptors. Significant β-blockade occurs at oral doses as low as 25 mg, and significant class III effects are seen only at daily doses of 160 mg and above [14]. It seems that the blockade of β-adrenoceptors contributes to antiseizure properties of sotalol, while decreased conductivity through KCNH_2_ potassium channels can lead to the exact opposite tendency. Therefore, the net effect on excitability may be the reason for inactivity of sotalol in the ECT test.

#### 4.1.5. Class IV

Diltiazem and verapamil are considered to be L-type calcium channel antagonists [72]; both of them present membrane-stabilizing activity [58]. Diltiazem, in contrast to verapamil, affected neither the electroconvulsive threshold nor the action of classical antiepileptics against the MES test. The main reason for this is a very low permeability of diltiazem through the blood–brain barrier [38,51,73].

### 4.2. Probable Mechanisms of Pharmacodynamic Interactions between Antiarrhythmic and Antiepileptic Drugs

According to Deckers et al. [74], if the two drugs applied in combination have different mechanisms of action, synergistic interactions between them are more likely; otherwise, additivity probably occurs. Therefore, in the case of positive interaction between two drugs, their mechanisms of action should be analyzed. Mechanisms of action of antiepileptic drugs have been presented in Table 5.

Antiseizure drugs taken into consideration in this review are: valproate, carbamazepine, phenytoin, phenobarbital, diazepam, oxcarbazepine, topiramate, pregabalin, lamotrigine, and lacosamide. The leading action of valproate, carbamazepine, phenytoin, oxcarbazepine, and lamotrigine is a blockade of VDSCs. Carbamazepine, phenytoin, lamotrigine, and oxcarbazepine block VDSCs during high-frequency discharges, but in therapeutic concentrations, they have no effect on physiological synaptic transmission [75]. Moreover, phenytoin, lamotrigine, and valproate also inhibit the persistent sodium currents [76]. Phenytoin, carbamazepine, oxcarbazepine, lamotrigine, and eslicarbazepine block VDSCs in the nonconducting fast inactivated state. On the other hand, lacosamide was shown to act on VDSCs channels during the slow activation phase [50,75,77]. Blocking of VDSCs in their inactivated conformation stabilizes this inactive form and prevents the return of the channel to the active state. In light of the above, phenytoin and carbamazepine seem to have very similar mechanisms of action on VDSCs. Both drugs limit high-frequency repetitive firing in a use-dependent manner, i.e., the higher the frequency of channel openings, the better the inhibitory effect of phenytoin and carbamazepine. Inhibiting high-frequency repetitive firing correlates well with a slower reactivation of VDSCs. Interestingly, both antiepileptics also exhibit membrane-stabilizing properties. On the other hand, phenytoin and carbamazepine at therapeutic doses affect neither GABA-ergic nor glutamatergic neurotransmission. However, there are also subtle differences between their action on VDSCs. Phenytoin requires a prolonged depolarization (around 100 ms) to start its action on sodium currents. Much shorter depolarization is enough for carbamazepine [58,75].

Regarding VDSCs subtypes located within α-subunit, valproate, carbamazepine, oxcarbazepine, lamotrigine, and topiramate block receptor subpopulations from Nav1.1 to Nav1.9. Phenytoin inhibits currents conducted through Nav1.1, 1.2, 1.5, and 1.8, while lacosamide shows an affinity for Nav1.3, 1.7, and 1.8 (data from DrugBank, https://go.drugbank.com).

Numerous antiepileptic drugs present more complex mechanisms of action. For instance, carbamazepine is an agonist of adenosine A_1_ and A_2_ receptors. It also blocks L-type voltage-dependent calcium channels (VDCCs) [75,78]. Anticonvulsant effects of oxcarbazepine may to some extent result from the enhancement of the outward potassium currents and blockade of high-voltage activated calcium channels of N/P and R types [78]. Interestingly, the blockade of N or P/Q channels inhibits the presynaptic release of excitatory amino acids [75,79].

In addition to the limitation of high-frequency repetitive firing, lamotrigine reduces the synaptic release of excitatory amino acids. In vitro studies revealed that lamotrigine enhances currents conducted through HCN channels and increases activity of GABA-ergic interneurons [75,80].

Valproate elevates GABA concentrations in the brain through three mechanisms: 1. increased synthesis of GABA by activating glutamic acid decarboxylase, a GABA synthesizing enzyme; 2. the increased potassium-induced release of GABA to the synaptic cleft; and 3. decreased activity of GABA-transaminase, an enzyme catalyzing GABA degradation. Moreover, valproate activates potassium conductance and inhibits low-threshold T-type calcium channels. The latter is crucial in inhibiting absence seizures [77,79,81,82].

Additionally, topiramate shows a complex mode of action. In addition to blocking VDSCs, the drug reduces excitatory neurotransmission through a negative modulatory effect on calcium-permeable AMPA/kainate receptors; potentiates GABA-mediated inhibitory neurotransmission through binding to a novel site within the GABA_A_ receptor complex; inhibits neuronal L-type high-voltage-activated calcium channels; weakly inhibits carbonic anhydrase; and activates potassium currents [75,79,83].

Phenobarbital is an agonist of the barbiturate recognition site, while diazepam binds to the benzodiazepine site within the GABA_A_ receptor complex. Barbiturates increase the time of GABA_A_-receptor-dependent chloride channel openings. In contrast, benzodiazepines increase the frequency but not the conductance or the time of opening of GABA_A_-related chloride channels. In addition, barbiturates can block VDSCs (but at a 10-times higher concentration than carbamazepine or phenytoin), activate voltage-dependent potassium channels, and inhibit AMPA-related glutamatergic transmission. Regarding benzodiazepines, other mechanisms of action include inhibition of adenosine uptake and, at higher doses, blocking VDSCs and voltage-dependent calcium channels [75,79,81].

Pregabalin acts presynaptically by binding to the alpha2-delta auxiliary subunit of voltage-dependent calcium channels. The drug reduces the calcium release, and, in consequence, several neurotransmitters, such as glutamate, substance P, and norepinephrine. Moreover, pregabalin blocks high-voltage-dependent calcium channels [75,79,84,85].

Lacosamide, in addition to its effect on slow activation of sodium channels, acts as an antagonist of the glycine-binding site on NMDA receptors [75].

Careful analysis of the combined treatment with antiarrhythmic and antiepileptic drugs (Table 1, Table 2 and Table 3) supports the widespread opinion that newer antiepileptic drugs are less likely to interact with other medications. Among the antiarrhytmic drugs, propafenone and mexiletine were the most active in interactions with antiseizure drugs. Both antiarrhythmic drugs block Nav1.5 currents and KCNH_2_ receptors and present membrane-stabilizing activity. Propafenone potentiated the antielectroshock action of valproate (*p* < 0.001), carbamazepine (*p* < 0.001), phenytoin (*p* < 0.01), phenobarbital (*p* < 0.01), pregabalin (*p* < 0.01), and topiramate (*p* < 0.05), but not that of lamotrigine [22,23]. Even considering that increased action of valproate, carbamazepine, and pregabalin was due in part to the increased brain concentration of the two drugs, it seems to be clear that the strongest interactions were observed between propafenone and antiepileptics blocking sodium channels. All of them block a wide range of sodium channels from Nav1.1 to Nav1.9. One could assume that in the case of identified drug-drug interactions it comes down to complementary effects of drug components on sodium channels. A reasoning problem arises with lamotrigine, which also blocks the whole range of sodium channels, and pregabalin reducing the release of excitatory amino acids to the synaptic cleft. At the present level of knowledge, the reason for the lack of interaction between the two antiepileptics with propafenone remains unclear.

In the case of mexiletine (Table 3), isobolographic analysis showed synergism with topiramate and pregabalin, which was independent of mexiletine-induced changes in brain levels of both antiepileptics. Surprisingly, a tendency towards antagonism was revealed for the combination of mexiletine and valproate in proportions of 1:1 and 3:1. Although the lowered brain concentration of valproate could be in part responsible for this result in the 1:1 proportion, no pharmacokinetic events were observed for the fixed ratios of 3:1 [24,25]. It can be suggested that a synergistic interaction occurred between mexiletine, a Nav1.5 and KCNH_2_ antagonist and two antiepileptic drugs with different mexiletine mechanisms of action. Additivity was observed between the antiarrhythmic drug and antiepileptics blocking VDSCs. However, this line of reasoning does not explain why additivity, not synergism, was shown in the case of phenobarbital (a strong enhancer of GABA-ergic transmission). Additionally, the reasons for the antagonism between mexiletine and valproate (in the 3:1 proportion) seem to be unclear. In this case, mexiletine did not lower the brain concentration of valproate.

Ivabradine, an inhibitor of HCN_2_ channels, enhanced the antielectroshock action of valproate but decreased that of phenytoin and lamotrigine. Pharmacokinetic interactions were only relevant in the case of phenytoin [20,21]. It is not clear why ivabradine exhibited the opposite effect on antiepileptic drugs, whose main mechanism of action is sodium channel blocking. The observed decrease in the anticonvulsant potency of lamotrigine in the presence of ivabradine could be explained by the common action of the two drugs on HCN channels [80]. Ivabradine potentially has a higher affinity for HCN channels and does not allow lamotrigine to express its full antiseizure properties.

Numerous β-blockers were examined in the MES test [28,30]. Their pharmacological properties are gathered in Table 4. Propranolol, pindolol, acebutolol, metoprolol, alprenolol, and nebivolol interacted with antiepileptic drugs. Propranolol, metoprolol, and acebutolol enhanced the action of valproate in the MES test. Furthermore, propranolol, pindolol, and alprenolol potentiated the antielectroshock effect of phenobarbital. Propranolol and metoprolol enhanced the action of diazepam. This indicates that β-blockers readily interact with antiepileptics enhancing GABA-ergic neurotransmission [30]. The main property of all β-receptor antagonists interacting with antiepileptic drugs is their membrane-stabilizing activity. The same can be said about nevivolol, but this drug behaved differently in the MES test, decreasing the action of carbamazepine. At present, there is no plausible explanation for this phenomenon.

Amiodarone potentiated the antielectroshock action of carbamazepine, oxcarbazepine, and pregabalin, whereas dronedarone potentiated that of lamotrigine. Surprisingly, dronedarone diminished the effect of phenytoin in the MES test in mice. The two pure multichannel blockers did not produce any pharmacokinetic interactions [32,33,34,35]. Surprisingly, amiodarone and dronedarone, although structurally related to each other, enhanced the action of different antiepileptic drugs. Additionally, processes underlying the dronedarone-induced decrease in the action of phenytoin remain incomprehensible. It should be remembered that the antielectroshock effect of phenytoin was not influenced by amiodarone and increased by sotalol. Even subtle differences in the mechanism of action of the component drugs are probably important for the resultant interactions between them. Undoubtedly, further molecular studies are necessary to explain the whole background of the above findings.

The effect of either valproate or phenytoin was potentiated by sotalol [36]. Additionally, these interactions seem to be pharmacodynamic. The interaction observed between sotalol and valproate could possibly be due to β-adrenolytic properties of sotalol, as pure β-blockers also enhanced the action of valproate.

Class IV antiarrhythmics, verapamil and diltiazem, block L-type calcium channels. In addition, verapamil moderately inhibits N, P, Q, and T calcium currents. It also slows conduction through KCNH_2_ channels and presents α1 antagonistic activity (DrugBank, https://go.drugbank.com). Because verapamil does not pass through the blood–brain barrier, it was ineffective against electrically induced seizures in mice. In contrast, diltiazem increased the ECT and enhanced the antielectroshock action of carbamazepine, phenytoin, and topiramate [38,40]. This effect may result from the complementary blockade of calcium and sodium channels. It does not explain, however, why diltiazem did not affect the action of other antiepileptic drugs inhibiting sodium currents.

### 4.3. Pharmacokinetic Interactions between Antiepileptic and Antiarrhythmic Drugs

In clinical conditions, plasma concentrations of carbamazepine, lacosamide, diazepam, and phenytoin can be increased by amiodarone, verapamil, diltiazem, pindolol, metoprolol, propranolol, and timolol. Worsening of undesired effects may necessitate the dose adjustment of antiepileptic drugs. Class 0, III, and IV antiarrhythmic drugs usually inhibit the metabolism of antiepileptics. However, some β-blockers, particularly metoprolol, can decrease excretion of carbamazepine or topiramate. On the other hand, antiepileptic drugs may affect plasma concentrations of amiodarone, dronedarone, nebivolol, propranolol, mexiletine, and ivabradine. In the majority of cases, the concentrations of antiarrhythmic drugs decrease, which may make the therapy ineffective. This happens particularly when antiarrhythmics are coadministered with carbamazepine, oxcarbazepine, phenytoin, and phenobarbital, the drugs known to increase activity of the hepatic cytochrome P450 enzymes. Among them, CYP3A4 is responsible for the metabolism of the largest number of clinically used drugs. Conversely, valproate can inhibit cytochrome P450 enzyme activity and increase plasma concentrations (and possibly adverse effects) of, e.g., amiodarone, mexiletine, verapamil, propranolol, and nebivolol (DrugBank, https://go.drugbank.com).

Interestingly, none of the above-mentioned interactions were confirmed in experimental studies discussed in this review. This supports the widely held opinion that pharmacokinetic interactions in animals do not translate into clinical practice.

### 4.4. Considerations Resulting from Analyzed Data

According to Raegan-Shaw et al. [86], maximal single doses of antiarrhythmic drugs used in patients have been converted to mouse doses (in mg/kg) and presented in Table 6 and Table 7. This allows us to find out whether doses applied in experimental studies are comparable and able to be interpolated to those used in clinical practice.

It appears that only diltiazem increased the ECT at doses comparable to human ranges; remaining antiarrhythmics affected this parameter at much higher doses than those applied in patients. More antiarrhythmics interacted with antiepileptic drugs at dosages close to clinical ranges, including propafenone, dronedarone, diltiazem, propranolol, metoprolol, alprenolol, and ivabradine. This may be of particular clinical importance in the case of dronedarone, ivabradine, and nebivolol. Dronedarone diminished the action of phenytoin, and ivabradine decreased the effect of phenytoin and lamotrigine, while nebivolol reduced the anti-MES action of carbamazepine.

In general, only some antiarrhythmic drugs exhibited their own anti-MES action or increased the ECT in mice. Moreover, in the case of propranolol and diltiazem, the available data remain inconsistent. The antielectroshock action of antiepileptic drugs was enhanced by all four classes of antiarrhythmic drugs and ivabradine. However, propafenone and mexiletine, i.e., sodium channel blockers, were the most active in this respect. On the other hand, the three mostly potentiated antiepileptics were valproate, carbamazepine, and phenytoin, whose mechanism of action is based on sodium channel blockades. This may contradict the theory that synergism happens only when the component drugs differ in their mechanisms of action. However, it should be remembered that sodium channels can be blocked in distinct target points and in divergent physiological states. On the other hand, most antiarrhythmics interacting with antiepileptic drugs, including propafenone, mexiletine, diltiazem, propranolol, pindolol, metoprolol, acebutolol, alprenolol, and nebivolol, present membrane-stabilizing activity. Additionally, antiepileptic drugs, whose antiseizure action is most often affected (valproate, carbamazepine, phenytoin), stabilize the cell membrane [87].

Interestingly, “pure” class III antiarrhythmics, amiodarone and dronedarone, did not interact pharmacokinetically with antiepileptic drugs. Remaining antiarrhythmics and ivabradine were able to increase or decrease brain concentrations of antiepileptic drugs; however, no regularity was found.

## 5. Conclusions

Analysis of the available data showed that interactions between antiarrhythmic and antiepileptic drugs are most likely when drug components block sodium channels and have membrane-stabilizing properties. From the clinical point of view, the most relevant interactions seem to be those where the action of antiepileptic drugs is reduced. This applies to interactions between mexilenine and valproate, dronedarone and phenytoin, ivabradine and phenytoin, as well as nebivolol and carbamazepine. If these results are confirmed clinically, the combinations mentioned above should be avoided.

## Figures and Tables

**Figure 1 ijms-23-02891-f001:**
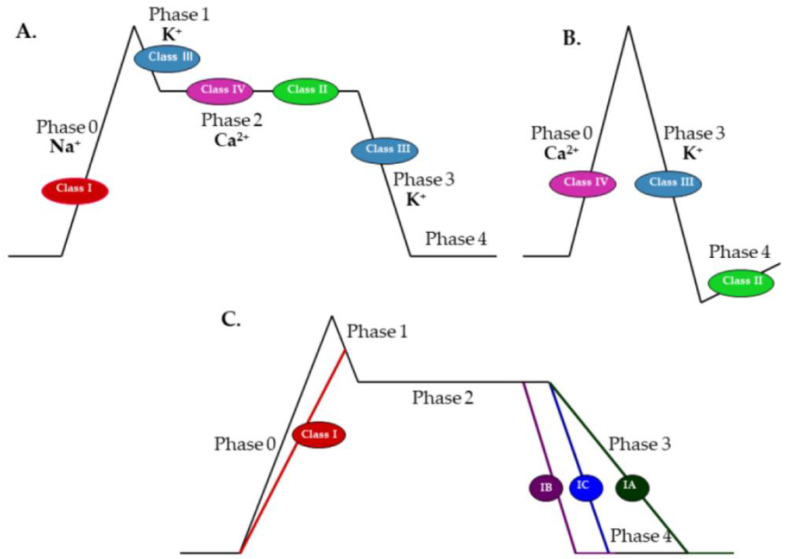
Effects of class I–IV antiarrhythmic drugs on phases of the action potential in: (**A**) cardiomyocytes; (**B**) sinoatrial and atrioventricular nodes; and (**C**) effects of subclasses IA, IB, and IC on phases of the action potential in cardiomyocytes (adapted and modified from https://www.ezmedlearning.com/blog/antiarrhythmics (accessed on 1 February 2022)).

**Table 1 ijms-23-02891-t001:** Effects of antiarrhythmic drugs on the antielectroshock action of antiepileptic drugs.

AADs	[Ref.]	Effect on ECT	Antiepileptic Drug Pharmacodynamic Effects	CNS Drug Concentration
Ivabradine	[19,20,21]	↑ ***	↑VPA *, ↓PHT **, ↓LTG *, ↔CBZ, ↔PB, ↔TPM, ↔PGB, ↔LSM	↓PHT *, not found for other AEDs
Propafenone	[22,23]	↑ **	↑VPA ***, ↑CBZ ***, ↑PB **, ↑OXC **, ↑TPM *, ↑PGB **, ↔LTG	↑VPA ***, ↑CBZ ***, ↑PGB *
Amiodarone	[32,33]	↔	↑CBZ **, ↑OXC *, ↑PGB ***, ↔VPA, ↔PHT, ↔PB, ↔TPM, ↔LTG	not found for any AEDs
Dronedarone	[34,35]	↑ ***	↑LTG *, ↓PHT *, ↔VPA, ↔CBZ, ↔PB, ↔TPM, ↔PGB, ↔LSM	not found for VPA, CBZ, PHT, PB, LTG, not tested for remaining AEDs
Sotalol	[36,37]	↔	↑VPA **, ↑PHT **, ↔CBZ, ↔PB, ↔OXC, ↔TPM, ↔PGB, ↔LTG	↑OXC ***, ↑TPM ***, ↓LTG **, not found for remaining AEDs
Propranolol	[26,28,29,30]	↑ _a_	↑VPA ***, ↑PB *, ↑DZP *, ↔CBZ, ↔PHT	not found for VPA, DZP, not tested for remaining AEDs
Acebutolol	[30]	↔	↑VPA **, ↔CBZ, ↔PHT, ↔DZP	not found for VPA, DZP, not tested for remaining AEDs
Metoprolol	[30]	↔	↑VPA ***, ↑DZP **, ↔CBZ, ↔PHT	not found for VPA and DZP, not tested for remaining AEDs
Atenolol	[30]	↔	↔VPA, ↔CBZ, ↔OXC, ↔DZP	not tested
Pindolol	[28]	↑ _a_	↑PB *	not tested
Alprenolol	[28]		↑PB *	not tested
Timolol	[29]	↔		not tested
Nebivolol	[31]	↔	↓CBZ **, ↔VPA, ↔PHT, ↔PB	↓VPA, not found for other AEDs
Verapamil	[38,39,40]	↔	↔VPA, ↔CBZ, ↔PB, ↔PHT, ↔TPM, ↔LTG	not found
Diltiazem	[38,39,40]	↔	↑CBZ **, ↑PHT ***, ↑TPM **, ↔VPA, ↔PB, ↔LTG	not found for CBZ, PHT, TPM, LTG, not tested for remaining AEDs

In certain studies, brain concentrations of antiepileptic drugs were measured only in the case of potentiation or attenuation of their antielectroshock action. CNS, central nervous system; ↑, increased electroconvulsive threshold/potentiated antielectroshock action/increased brain concentration; ↓, decreased electroconvulsive threshold/reduced antielectroshock action/decreased brain concentration; ↔, no effect on aforementioned parameters; AADs, antiarrhythmic drugs, AEDs, antiepileptic drugs; VPA, valproate; CBZ, carbamazepine; PB, phenobarbital; PHT, phenytoin; OXC, oxcarbazepine; TPM, topiramate; PGB, pregabalin; LTG, lamotrigine; LSM, lacosamide; _a_, active in the MES test at a very high dose (50 mg/kg); *, *p* < 0.05; **, *p* < 0.01, ***, *p* < 0.001 vs. the control (vehicle-treated mice).

**Table 2 ijms-23-02891-t002:** Effects of antiarrhythmic drugs on the antielectroshock action of individual antiepileptic drugs.

Antiepileptic Drugs	Antiarrhythmic Drugs Affecting the Antielectroshock Action of a Given Antiepileptic Drug
Valproate	↑propafenone, sotalol, propranolol, acebutolol, metoprolol, ivabradine
Carbamazepine	↑propafenone, amiodarone, diltiazem, ↓nebivolol
Phenytoin	↑propafenone, sotalol, diltiazem, ↓dronedarone, ivabradine
Phenobarbital	↑propafenone, propranolol, alprenolol, pindolol
Oxcarbazepine	↑propafenone, amiodarone
Topiramate	↑propafenone, diltiazem
Pregabalin	↑propafenone, amiodarone
Lamotrigine	↑dronedarone, ↓ivabradine
Diazepam	↑propranolol, metoprolol

↑, increased antielectroshock action of a given antiepileptic drug; ↓, decreased antielectroshock action of a given antiepileptic drug.

**Table 3 ijms-23-02891-t003:** Isobolographic interactions between mexiletine and antiepileptic drugs in the MES test [24,25].

MXL/AED Ratio	VPA	CBZ	PHT	PB	OXC	TPM	PGB	LTG
1:3	Add	Add	Add	Add	Add _ph↓_	S	Add _ph↓_	Add
1:1	Ant _ph↓_	Add	Add	Add	Add	S _ph↓_	S _ph↑_	Add
3:1	Ant	Add	Add	Add	Add	S	S _ph↓_	Add

MXL, mexiletine; AED, antiepileptic drug; VPA, valproate; CBZ, carbamazepine; PHT, phenytoin; PB, phenobarbital; OXC, oxcarbazepine; TPM, topiramate; PGB, pregabalin; LTG, lamotrigine; Add, additivity; S, synergism; Ant, antagonism; _ph↓_, decrease in the brain level of a certain AED, _ph↑_, increase in the brain level of a certain AED.

**Table 4 ijms-23-02891-t004:** Pharmacological properties of β-blockers.

Drug	Lipophilicity	β_1_-Selectivity	ISA	MSA	β_1_-BP
PROP	+++	0	0	+	1.0
PIND	+++	0	++	+	6.0
METO	+++	++	0	+	1.0
ATEN	+	+	0	0	1.0
ACEB	++	+	+	+	0.3
ALPR	+	0	0	0	0.6
TIM	+++	0	0	0	0.6
NEB	++	+++	0	0	10.0
SOT	+	0	0	0	0.3

PROP, propranolol; PIND, pindolol; METO, metoprolol; ATEN, atenolol; ACEB, acebutolol; ALPR, alprenolol; TIM, timolol; NEB, nebivolol; SOT, sotalol; ISA, intrinsic sympathomimetic activity; MSA, membrane-stabilizing activity; BP, blockade potency; +, weak action; ++, moderate action, +++, strong action; 0, no significant action(for review see [63,64]).

**Table 5 ijms-23-02891-t005:** Antiepileptic drugs—mechanisms of action at therapeutic concentrations.

Mechanism	VPA	CBZ	PHT	OXC	LTG	PB	DZP	TPM	PGB	LCS
Na channels	++	++	++	++	++	+	0	++	0	++
Ca channels	+ ^(T)^	+ ^(L)^	0	+ ^(N, P)^	+ ^(N, P, Q, R, T)^	0	+ ^(L)^	+ ^(N, P, Q)^	0	0
K channels	+	0	0	0	+	+	0	+	0	0
GABA	+	0	0	0	+	++	++	+	0	0
GLU	0	0	0	0	+	+	0	++	+	+
HCN	0	0	0	0	+	0	0	0	0	0
Adenosine	0	0	0	0	+	0	+	0	0	0

VPA, valproate; CBZ, carbamazepine; PHT, phenytoin; OXC, oxcarbazepine; LTG, lamotrigine; PB, phenobarbital; DZP, diazepam; TPM, topiramate; PGB, pregabalin; LCS, lacosamide; ++, strong action; +, weak action; 0, nonsignificant action; ^L^, ^N^, ^P^, ^Q^, ^R^, ^T^, types of voltage-dependent calcium channels (VDCC).

**Table 6 ijms-23-02891-t006:** Doses (in mg/kg) of antiarrhythmic drugs used in the ECT test and combinations with AEDs in the MES test.

AADs	ECT	VPA	CBZ	PHT	PB	OXC	TPM	PGB	LTG	DZP
Propafenone	60–90	20	2.5	50	30	40	50	50	-	-
Amiodarone	-	-	75	-	-	100	-	87.5	-	-
Dronedarone	75–100	-	-	50	-	-	-	-	50	-
Sotalol	-	60	-	80	-	-	-	-	-	-
Diltiazem	2.5–5.0	0.62	0.62	-	-	-	-	-	-	-
Propranolol	50	5	-	-	5	-	-	-	-	5
Acebutolol	-	100	-	-	-	-	-	-	-	-
Metoprolol	-	50	-	-	-	-	-	-	-	50
Pindolol	-	-	-	-	5	-	-	-	-	-
Alprenolol	-	-	-	-	5	-	-	-	-	-
Nebivolol	-	-	15	-	-	-	-	-	-	-
Ivabradine	15–20	10	-	10	-	-	-	-	-	-

ECT, electroconvulsive threshold test; MES, maximal electroshock test; AEDs, antiepileptic drugs.

**Table 7 ijms-23-02891-t007:** Doses of antiarrhythmic drugs (in mg/kg) used in the ECT and/or MES tests in mice, single maximal therapeutic dose in humans, and the converted single maximal therapeutic dose in mice.

Antiarrhythmic Drugs	Min Mouse D	Max Mouse D	Max Therapeutic Human D	Max Therapeutic Human D Converted to Mouse D
Propafenone	2.5	50	300	53.60
Amiodarone	75	100	200	35.71
Dronedarone	50	50	400	71.43
Sotalol	60	80	240	42.86
Diltiazem	0.62	0.62	90	16.07
Propranolol	5	5	100	17.86
Acebutolol	100	100	400	71.43
Metoprolol	50	50	400	71.43
Pindolol	5	5	15	2.68
Alprenolol	5	5	200	35.71
Nebivolol	15	15	5	0.89
Ivabradine	10	10	5	0.89

Min mouse D, Max mouse D (mg/kg), minimal and maximal dose of antiarrhythmic drugs used in the electroconvulsive threshold (ECT) test and maximal electroshock (MES) in mice; max therapeutic human D (mg), maximal single therapeutic dose in humans; max therapeutic human D converted to mouse D, maximal therapeutic human dose converted to theoretical maximal therapeutic dose in mice according to Raegan-Shaw et al. [86].

## Data Availability

The data presented in this study are available on request from the corresponding author.

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
