# Peer review of "Effects of Antiarrhythmic Drugs on Antiepileptic Drug Action—A Critical Review of Experimental Findings"

_ijms, 2022, doi:10.3390/ijms23052891_

Round 1

Reviewer 1 Report

In this review article, the author tried to analyze the influence of antiarrhythmic drugs and ivabradine on the effectiveness of anti-seizure treatment. The author concluded that interactions between antiarrhythmic and antiepileptic drugs are the most likely, when drug components block sodium channels and have membrane stabilizing properties.

Comments

This is an interesting review article. The reviewer has some concerns as follows:

  1. In the Abstract:

(1) line 11, please explain what is “i.a.”?

(2) line 12, there is a grammar error for “The aim of this review was analysis of such…”.

  1. In the Introduction:

(1) line 35-36, the description for “It is still the most commonly used animal model of generalized tonic-clonic seizures in humans.” is confusing and unconvincing.

(2) line 68, the description for “According to Vaughan Williams…” is confusing. It needs a reference.

(3) line 96-101, the sentence needs to be revised.

(4) line 121-122, there is a grammar error for this sentence.

  1. In the Selection of literature:

(1) line 134-135, the sentence for “Out of (Medline) and (Science Direct) results, 23 articles covering more than 30 year-research were selected as the most relevant.” Is confusing and needs to be revised.

  1. In the Antiarrhythmic drugs have different effects on tonic-clonic convulsions:

(1) line 139-140, there is a grammar error for “with results. Result were considered…”.

(2) line 142-149, the references need to be cited for these descriptions.

(3) line 148-149, the description for “…and remaining antiepileptic drugs used in this study.” is confusing and needs to be revised.

(4) line 151, the description for “…the interaction between ivabradine and valproate seems pharmacodynamic in nature…” is confusing. What meaning is “pharmacodynamic in nature”?

(5) line 160-162, this sentence seems incomplete. The action of carbamazepine was potentiated by what?

(6) line 172, the description for “seem to be pharmacodynamic in nature” is confusing and needs to be revised.

(7) line 175-187, the references need to be cited for these descriptions.

(8) line 204-206, the sentence for “In another study, propranolol, acebutolol, metoprolol and atenolol given at doses not affecting the ECT influenced the antiseizure action of some antiepileptic drugs in the MES test.” is confusing and can be revised.

(9) line 230-236, the references need to be cited for these descriptions.

(10) line 246-248, this sentence can be revised. What is meaning for “are most likely pharmacodynamic in nature”.

(11) In Table 2, line 324, “diltiazem ↓ dronedarone” change to “diltiazem, ↓ dronedarone”; line 329, “↑ dronedarone; ↓ ivabradine” change to “↑ dronedarone, ↓ ivabradine”.

  1. In the Discussion:

(1) line 365-366, there is a grammar error for “Inhibition of potassium and calcium currents appear to be additional factors…”.

(2) line 385, there is a grammar error for “Then, fast and slow inactivation turn the channel into a non-conducting…”.

(3) line 425-427, the description for “Among 8 β-blockers compared in this re-425 view, only propranolol and pindolol, given in high doses, were active against MES in 426 mice.” is confusing and can be revised.

(4) line 477, this sentence seems incomplete and has a grammar error for “Table 2. potassium channels (KCNH2). KCNH2 channels suppresses influx of K+ ions…”.

(5) line 635, this sentence for “Also unclear remain reasons of antagonism between mexiletine and valproate.” seems incomplete and needs to be revised.

(6) line 667-668, this sentence for “Also these interactions were pharmacodynamic in nature.” is confusing and can be revised.

(7) line 691, “phenytoin, ivabradine” change to “phenytoin and ivabradine”.

(8) line 695, there is a grammar error for this sentence.

(9) line 696-697, there is a grammar error for this senctence.

(10) The Table 7 can be reorganized.

Author Response

Reply to the Reviewer 1

I would like to thank you very much for your insightful review. It made the language correction much easier. Additionally, the whole text has been corrected by a professional English translator. Hopefully, the current version is more convincing.

Detailed response:

In the Abstract:

(1) line 11, please explain what is “i.a.”?

Inter alia (i.a.) is Latin for "among others". In the corrected form I used the English phrase.

(2) line 12, there is a grammar error for “The aim of this review was analysis of such…”.

This sentence has been rewritten.

  1. In the Introduction:

(1) line 35-36, the description for “It is still the most commonly used animal model of generalized tonic-clonic seizures in humans.” is confusing and unconvincing.

This sentence has been rewritten.

(2) line 68, the description for “According to Vaughan Williams…” is confusing. It needs a reference.

The reference [4}] has been added. Actually, references 4,14, and 15 (line 91) apply to the entire paragraph. The Vaughan Williams classification was introduced more than 50 years ago. This classification  listed in all textbooks on cardiology and pharmacology and it is as obvious as, for example, Newton's laws in physics. To the best of my knowledge, Miles Vaughan Williams did not publish his classification in any separate article.

(3) line 96-101, the sentence needs to be revised.

This sentence has been revised.

(4) line 121-122, there is a grammar error for this sentence.

This sentence has been rewritten.

  1. In the Selection of literature:

(1) line 134-135, the sentence for “Out of (Medline) and (Science Direct) results, 23 articles covering more than 30 year-research were selected as the most relevant.” Is confusing and needs to be revised.

This sentence has been revised.

  1. In the Antiarrhythmic drugs have different effects on tonic-clonic convulsions:

(1) line 139-140, there is a grammar error for “with results. Result were considered…”.

This sentence has been rewritten.

(2) line 142-149, the references need to be cited for these descriptions.

The whole paragraph is assigned to references [19-21].

(3) line 148-149, the description for “…and remaining antiepileptic drugs used in this study.” is confusing and needs to be revised.

This sentence has been revised.

(4) line 151, the description for “…the interaction between ivabradine and valproate seems pharmacodynamic in nature…” is confusing. What meaning is “pharmacodynamic in nature”?

This sentence has been rewritten. The phrase „in nature”  has been deleted.

(5) line 160-162, this sentence seems incomplete. The action of carbamazepine was potentiated by what?

This confusing sentence has also been rewritten.

(6) line 172, the description for “seem to be pharmacodynamic in nature” is confusing and needs to be revised.

The phrase „in nature” has been deleted.

(7) line 175-187, the references need to be cited for these descriptions.

The reference [25] applies to the whole paragraph.

(8) line 204-206, the sentence for “In another study, propranolol, acebutolol, metoprolol and atenolol given at doses not affecting the ECT influenced the antiseizure action of some antiepileptic drugs in the MES test.” is confusing and can be revised.

This sentence has been revised.

(9) line 230-236, the references need to be cited for these descriptions.

The reference [35] applies to the whole paragraph.

(10) line 246-248, this sentence can be revised. What is meaning for “are most likely pharmacodynamic in nature”.

The phrase „in nature” has been deleted.

(11) In Table 2, line 324, “diltiazem ↓ dronedarone” change to “diltiazem, ↓ dronedarone”; line 329, “↑ dronedarone; ↓ ivabradine” change to “↑ dronedarone, ↓ ivabradine”.

Everything has been corrected.

  1. In the Discussion:

(1) line 365-366, there is a grammar error for “Inhibition of potassium and calcium currents appear to be additional factors…”.

This sentence has been corrected.

(2) line 385, there is a grammar error for “Then, fast and slow inactivation turn the channel into a non-conducting…”.

This sentence has been corrected.

(3) line 425-427, the description for “Among 8 β-blockers compared in this re-425 view, only propranolol and pindolol, given in high doses, were active against MES in 426 mice.” is confusing and can be revised.

This sentence has been revised.

(4) line 477, this sentence seems incomplete and has a grammar error for “Table 2. potassium channels (KCNH2). KCNH2 channels suppresses influx of K+ ions…”.

The whole beginning of this paragraph has been rewritten.

(5) line 635, this sentence for “Also unclear remain reasons of antagonism between mexiletine and valproate.” seems incomplete and needs to be revised.

This sentence has been revised.

(6) line 667-668, this sentence for “Also these interactions were pharmacodynamic in nature.” is confusing and can be revised.

Also this sentence has been revised.

(7) line 691, “phenytoin, ivabradine” change to “phenytoin and ivabradine”.

It has been corrected.

(8) line 695, there is a grammar error for this sentence.

This error has been corrected.

(9) line 696-697, there is a grammar error for this senctence.

It has also been corrected.

(10) The Table 7 can be reorganized.

I tried to correct the Table 7 to make it more readable. If I have not lived up to expectations, please give me some suggestion.

Reviewer 2 Report

This is a good review summarizing the current knowledge about interactions between antiarrhythmic and antiepileptic drugs.

However, I have some comments and suggestions:

Title should be more precise, i.e. as authors only present information about effects of antiarrhythmic drugs on antiepileptic drugs (mostly pharmacodynamic) in experimental models  – e.g. Effects of antiarrhythmic drugs on antiepileptic drug actions - a critical review of experimental findings

Abstract:

instead drug interactions, a term drug-drug interactions should be used (as it is more precise in the context of the review title).

Introduction:

Line 77: is β-adrenergic inhibitors; should be β-adrenergic blockers

Table 1

Effect on antiepileptic drugs – more correctly should be – Antiepileptic drug pharmacodynamics effects

Pharmacokinetics - more correctly should be – CNS drug concentration

Table 2

Antidepressant drugs affecting the antielectroshock action of a given antiepileptic drug – should be replaced by - Antiarrhythmic drugs (line 320)

Table 3

Heading – spelling error in “antiepileptic” word

Line 477 – table 2 – probably formatting error

If the author expand the information about impact of antiepileptic drugs on antiarrhythmic drug actions, the title should be modified accordingly.

In discussion potential pharmacokinetic interactions mechanisms should be stated, e.g. enzyme activity (amiodarone-phenytoin).

Author Response

Reply to the Reviewer 2

Thank you very much for your kind review. All suggestions have been taken into account.

Detailed response:

Title should be more precise, i.e. as authors only present information about effects of antiarrhythmic drugs on antiepileptic drugs (mostly pharmacodynamic) in experimental models  – e.g. Effects of antiarrhythmic drugs on antiepileptic drug actions - a critical review of experimental findings

The title has been changes according to this suggestion.

 Abstract:

instead drug interactions, a term drug-drug interactions should be used (as it is more precise in the context of the review title).

This phrase has been corrected. 

Introduction:

Line 77: is β-adrenergic inhibitors; should be β-adrenergic blockers

Also this phrase has been revised.

Table 1

Effect on antiepileptic drugs – more correctly should be – Antiepileptic drug pharmacodynamics effects

Pharmacokinetics - more correctly should be – CNS drug concentration

The Table 1 headings have been changed.

Table 2

Antidepressant drugs affecting the antielectroshock action of a given antiepileptic drug – should be replaced by - Antiarrhythmic drugs (line 320)

Of course, this obvious mistake has been corrected.

Table 3

Heading – spelling error in “antiepileptic” word

The error has been corrected.

Line 477 – table 2 – probably formatting error

The beginning of thi paragraph has been entirely revised.

 If the author expand the information about impact of antiepileptic drugs on antiarrhythmic drug actions, the title should be modified accordingly.

Knowledge of the impact of antiepileptic drugs on antiarrhythmics is so vast that it may be considered as a topic for the next review. However, such a review would cover quite different animal models.

In discussion potential pharmacokinetic interactions mechanisms should be stated, e.g. enzyme activity (amiodarone-phenytoin).

Potential pharmacokinetic interactions were described in general on Pg 16 (para 4.3.)

Round 2

Reviewer 1 Report

This revised manuscript can be accepted. No further comments.